# The Shapes of Stellar Spectra

Carlos Allende Prieto [1,2]

1 Instituto de Astrofísica de Canarias, Vía Láctea S/N, E-38205 La Laguna, Tenerife, Spain; callende@iac.es;
Tel.: +34-922-605200
2 Departamento de Astrofísica, Universidad de La Laguna, E-38206 La Laguna, Tenerife, Spain

**Abstract:** Stellar atmospheres separate the hot and dense stellar interiors from the emptiness of space. Radiation escapes from the outermost layers of a star, carrying direct physical information. Underneath the atmosphere, the very high opacity keeps radiation thermalized and resembling a black body with the local temperature. In the atmosphere the opacity drops, and radiative energy leaks out, which is redistributed in wavelength according to the physical processes by which matter and radiation interact, in particular photoionization. In this article, I will evaluate the role of photoionization in shaping the stellar energy distribution of stars. To that end, I employ simple, state-of-the-art plane-parallel model atmospheres and a spectral synthesis code, dissecting the effects of photoionization from different chemical elements and species, for stars of different masses in the range of 0.3 to 2 $M_\odot$. I examine and interpret the changes in the observed spectral energy distributions of the stars as a function of the atmospheric parameters. The photoionization of atomic hydrogen and $H^-$ are the most relevant contributors to the continuum opacity in the optical and near-infrared regions, while heavier elements become important in the ultraviolet region. In the spectra of the coolest stars (spectral types M and later), the continuum shape from photoionization is no longer recognizable due to the accumulation of lines, mainly from molecules. These facts have been known for a long time, but the calculations presented provide an updated quantitative evaluation and insight into the role of photoionization on the structure of stellar atmospheres.

**Keywords:** stars; stellar atmospheres; opacity

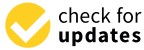



## 1. Introduction

Starlight escapes from the atmospheres of stars, which become optically thin at different heights depending on the wavelength. The main effect causing photon absorption in stellar atmospheres is photoionization. As a result, this process reshapes the spectral energy distribution of stars, which in deeper layers must resemble black bodies, and dictates the details of how their colors change with surface temperature (or mass). In what follows, the discussion will be focused on stars with masses between 0.3 and about 2 $M_\odot$. There are of course stars with less (down to the minimum mass for hydrogen burning, about 0.07 $M_\odot$), and more mass (probably up to $\sim 100$ $M_\odot$), but most stars are included in this range.

Photoionization in stellar atmospheres is mainly associated with hydrogen atoms and closely related ions, in particular $H^-$, which is a dominant source of absorption for solar-type stars. This is not surprising since 80% of the stellar mass is usually hydrogen, and the remaining is mostly helium, with less than 2% left for the rest of the elements. Hydrogen photoionization imprints a series of discontinuities on stellar spectra, related to the various atomic levels: the Lyman ($n = 1$), Balmer ($n = 2$), Paschem ($n = 3$), Brackett ($n = 4$), Pfund ($n = 5$), etc. series, visible at wavenumbers of $R_H/n^2$, where $R_H$ is the Rydberg constant, or wavelengths of 912, 3646, 8204, 14580, 22790 Å, etc. In thermodynamical equilibrium, the Saha equation shows that the ionization fraction for H is inversely proportional to the electron density, and therefore higher electron pressure, as present on a main-sequence star, burning hydrogen in its core, compared to an evolved giant star, leads to less ionization of H and more pronounced photoionization jumps.

For most stars, there is a competition between the opacity from photoionization of atomic H and H$^-$ at optical and infrared wavelenghts, and depending on this, the H series jumps become more or less prominent. In the infrared region, from $\lambda \sim 16{,}000$ Å onwards, free-free (inverse bremsstrahlung) typically overcomes bound-free (photoionization) for H$^-$. In the ultraviolet region, the situation becomes much more complex, since there are multiple, heavier elements that become important contributors to the opacity through ionization; usually carbon, sodium, magnesium, aluminum, silicon, and iron become very important at different wavelengths.

Reports from the 1970s to the 1990s claimed a significant mismatch between models and observations in the ultraviolet for the Sun, suggesting a missing opacity source. More recent studies (see, e.g., [1–3]) claimed to have identified the source as iron, at least in part associated with autoionizing lines missed in the earliest calculations.

Nevertheless, the far UV spectrum of solar-like stars is very hard to model, since the accumulated opacity makes photons to escape from the upper atmosphere (chromosphere, transition region, and corona), rather than the much-better-understood and easier-to-model photosphere. In those high layers, the low density, high temperatures, and the presence of magnetic fields complicate physical modeling. For moderately warmer stars, on the other hand, such as B-type stars, UV light escapes from deeper, photospheric, layers, which are simpler.

This paper, devoted to the role of photoionization in shaping stellar spectra, will first describe the data sources and codes used in our calculations. In Section 3, I dissect the various contributors to the opacity in the atmospheres of the Sun and other types of stars. Section 4 will test the models against selected high-quality spectrophotometric observations, giving us an idea of the realism of the models, as well as their limitations. Section 5 examines the sensitivity of the stellar continuum to changes in the main atmospheric parameters: surface effective temperature, surface gravity, and metallicity[1] (the fraction of heavy elements). The paper closes with a short summary and conclusions in Section 6.

## 2. Adopted Data

In the calculations below, we adopt classical 1D plane-parallel model atmospheres from the MARCS [4,5] and Kurucz ([3] and updates) grids—see also [6] for the most recent incarnations. The theory of stellar atmospheres is laid out in detail, for example, in the textbook [7].

The sources of atomic data adopted for the opacity and synthetic spectra computed are mostly those described in [8], with some updates. Photoionization cross-sections are from TOPBASE [9] for all the elements considered but iron, which are from [10,11]. Bound-free and free-free absorption was also considered for H$^-$, H$_2^+$, He$^-$, CH, OH, H$_2^-$, as well as collisionally induced opacity from H$_2$-H$_2$, H$_2$-He, H$_2$-H, and H-He. Rayleigh scattering on atomic hydrogen and helium, H$_2$, and the wings of Lyman alpha were also included.

Atomic line data are from the most recent files from Kurucz's website[2] updated with damping constants from [12,13]. Kurucz's website (and references therein) is also the source for the molecular line data, including H$_2$, CH, C$_2$, CN, CO, NH, OH, MgH, SiH, SiO, AlO, CaH, CaO, CrH, FeH, MgO, NaH, SiH, and VO, with the exception of Exomol data employed for TiO [14] and H$_2$O [15].

The actual calculations were performed with the latest version of the code Synspec [16,17]. The data and the code used are bundled with the Python wrapper Synple[3] [16], version 1.2.

## 3. Atmospheric Opacity

A star is a fairly independent entity, kept together by its own gravitational pull, and for most of its life producing its own energy through nuclear fusion in the core, where the temperature reaches tens of millions of degrees. Ionization is very high throughout the stellar interior up to the surface, where the temperature drops under $T \sim 10{,}000$ K, hydrogen becomes neutral, the electron density falls dramatically, and with it the material becomes transparent and radiation escapes.

The opacity at the stellar surface shapes the stellar spectrum, which is mainly due to the photoionization and inverse bremsstrahlung of the first few ionization stages of the most abundant elements, chiefly hydrogen and, in solar-type stars or cooler types, $H^-$ and other molecules (see, e.g., [18]).

The top panel of Figure 1 illustrates the run of temperature vs. density in the atmosphere of a solar-like star. The point marked with a circle is just above the layers where the temperature of the star matches its effective temperature ($T_{\rm eff}$, defined as that of a black body with the same radiative flux $F = \sigma T_{\rm eff}^4$, where $\sigma$ is the Stefan-Boltzmann constant, and from where the optical continuum is escaping. The bottom panel of the figure shows the total (blue) and photoionization (plus bremsstrahlung) (orange) opacity at the chosen point ($T = 5100$ K, $\rho = 1 \times 10^{-7}$ g cm$^{-3}$).

As we mentioned in the introduction, the smooth curve that dominates the continuum opacity between 4000 and 16,000 Å is due to the photoionization of $H^-$, while the rising continuum curve at longer wavelengths is due to $H^-$ bremsstrahlung. In the UV region, photoionization from abundant neutral and singly ionized elements is responsible for the ragged and rapidly increasing continuum opacity. For each ion, the total opacity is the sum of the contributions from multiple levels. The height over the continuum reached by many of the lines suggests that line opacity is dominant, but most of these lines are narrow, with an FWHM of a fraction of an Ångstrom, and the shape of the stellar spectrum is, at least in the optical and near-infrared regions, dominated by photoionization.

Figure 2 dissects the continuum opacity, for the same conditions in the solar atmosphere adopted in Figure 1, into the various atomic and molecular contributors. Neutral carbon, magnesium, aluminum, silicon, and iron show up as the most important absorbers in the UV region. However, while the important role of iron at $\lambda < 3000$ Å and magnesium at $\lambda < 2500$ Å is undeniable, the dramatic Increase in opacity, not only due to photoionization but also through the accumulation of transitions at these wavelengths, shifts the layers from which UV radiation escapes higher up, and the importance of some of the ionization edges is hard to assess in practice.

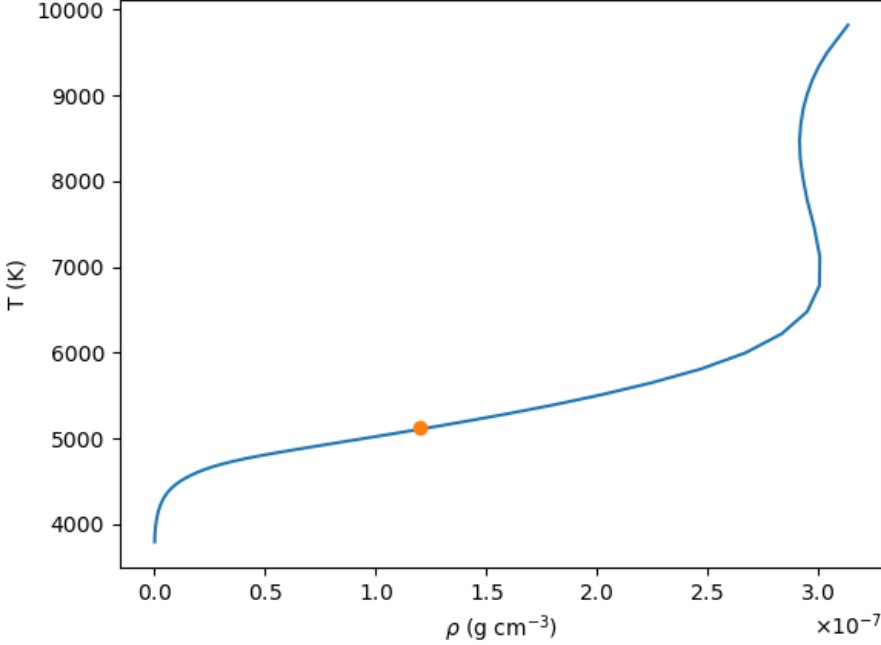

**Figure 1.** *Cont.*

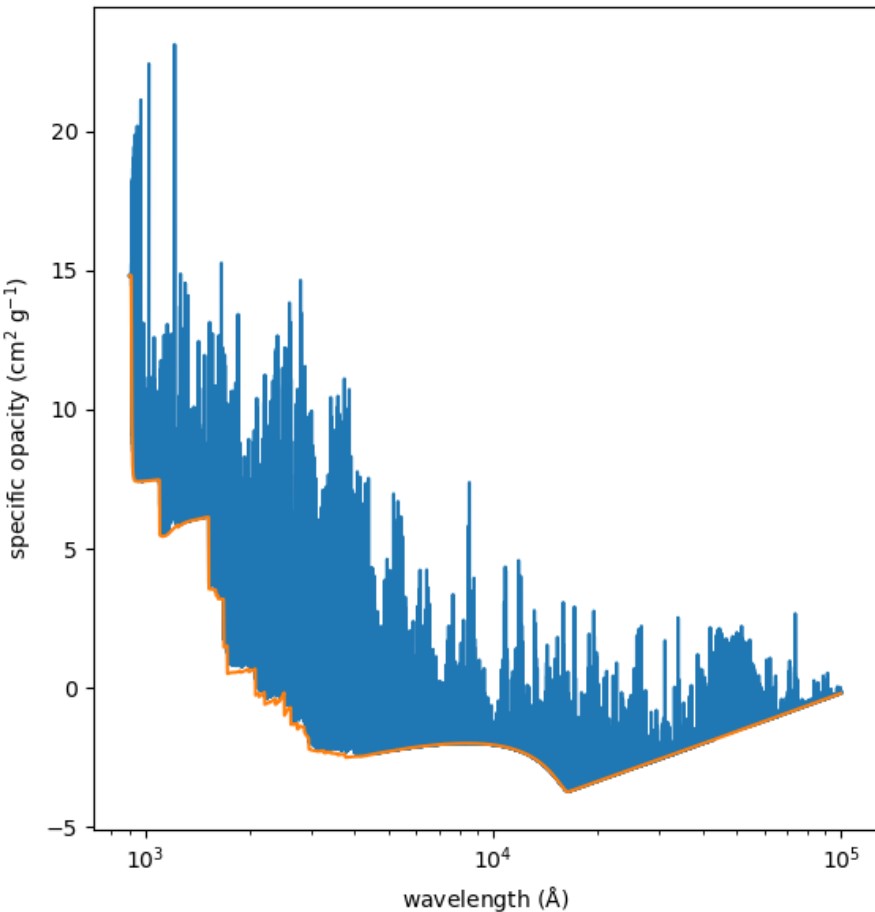

**Figure 1.** (**Upper panel**) Relationship between temperature and density in the atmosphere of a solar-like star ($T_{\rm eff} = 5777$ K, $\log g = 4.437$ with $g$ in cm s$^{-2}$, and the chemical abundances given in [19]) from a plane-parallel model in Local Thermodynamical Equilibrium. The indicated point is just slightly higher than optical depth unity, corresponding to $\rho \simeq 1.2 \times 10^{-7}$ g cm$^{-3}$ and $T \simeq 5100$ K. (**Lower panel**) Continuous (photoionization plus bremsstrahlung; orange) and total opacity for the point in the T-$\rho$ run marked in the top panel.

This picture may be incomplete, since the list of ions included in the calculations is not exhaustive, and has in practice been limited to those for which there are calculations available from the Opacity Project and the Iron Project. Furthermore, some of the opacity due to autoionization lines may be included twice as line transitions and resonances in the photoionization cross sections.

The agreement with observations (see Section 4) suggests the major contributors to opacity in stellar atmospheres have been identified, but there may be modestly or moderately important contributors missing. In a recent paper [20] it has been pointed out that the free-free opacity cross-section for negative positronium ions (the equivalent of H$^-$ for a positronium instead of a H atom) would be larger than that of H$^-$ in the infrared for the solar atmosphere. Nonetheless, the lack of knowledge of the abundance of positrons in the solar atmosphere makes it hard to assess how much relevance such a contribution would make to the total opacity.

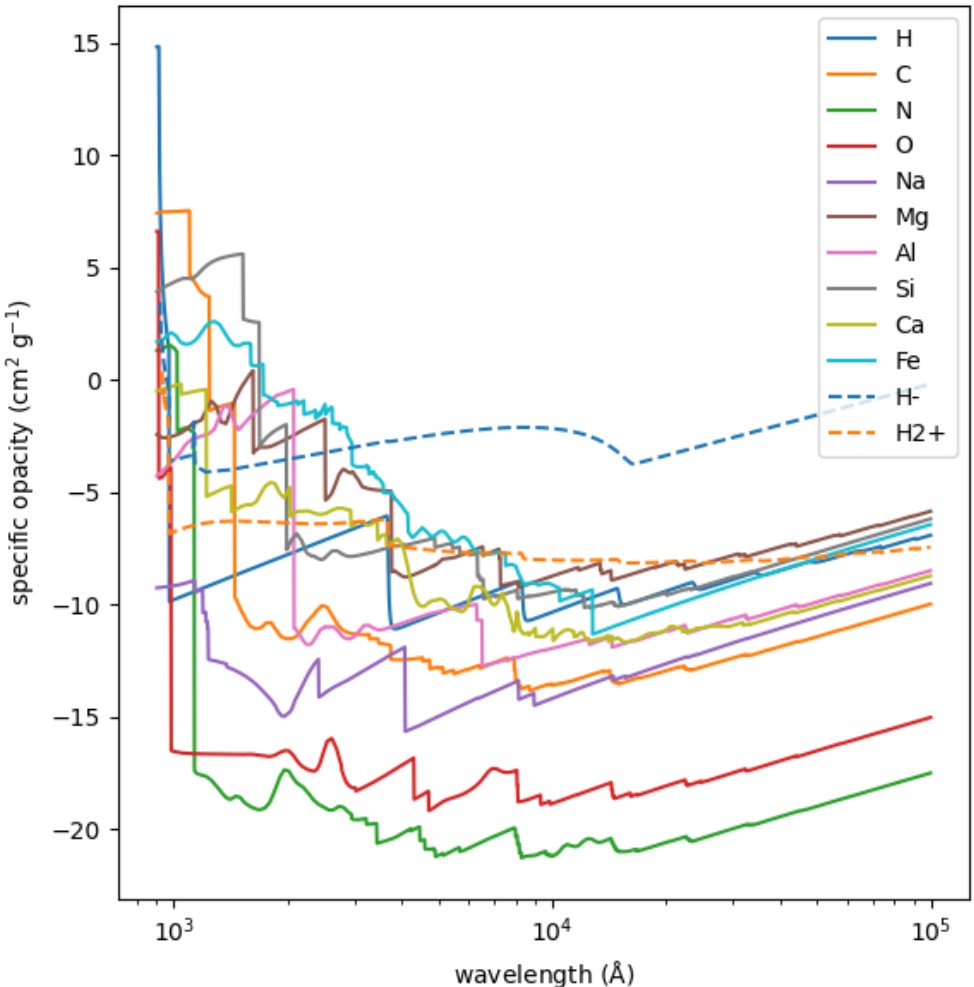

**Figure 2.** Bound-free Author Ok and free-free opacity associated with individual elements for $\rho \simeq 1.2 \times 10^{-7}$ g cm$^{-3}$ and $T \simeq 5100$ K. The continuous opacity for H$^-$ and H$_2^+$ is also shown.

Figure 3 repeats the information displayed in Figure 1 for the Sun (shown here in orange), but also includes a model for an A-type star $T_{\text{eff}} = 9800$ K) and a much cooler M-type star ($T_{\text{eff}} = 3500$ K). As before, we have chosen atmospheric depths representative of the regions where the optical continuum escapes. The lower panel shows the photoionization (and bremsstrahlung) opacity for the three stars with dashed lines. For the hot (A-type, blue) and cool stars (M-type, red), the total opacity, including lines, is also shown.

The continuum of the M-type star is, like for the Sun, shaped by H$^-$ photoionization between 4000 and 16,000 Å. H$^-$ free-free opacity dominates at longer wavelengths, and the photoionization of heavier elements (and electron scattering) is most relevant in the UV. However, line absorption severely blocks much of the continuum flux. The continuum of the warmer A-type star, on the other hand, shows the characteristic shape of the hydrogenic photoionization cross-section, proportional to $\lambda^3$ [21], with discontinuities corresponding to the minimum ionization energies for each of the $n = 1, 2, 3, \ldots$, etc. levels. Note that the H$^-$ bremsstrahlung opacity shares the same slope [22].

Line absorption is progressively reduced as the surface temperature of the star increases, due to the ionization of the main line absorbers, chiefly atomic iron. It is quite obvious in Figure 3 that the lines add sharp opacity peaks on top of the continuum for the A-type star, while the lower edge of the line absorption appears detached from the continuum opacity for the cooler M-type star, indicative of a massive accumulation of lines that overlap, mainly from molecules such as CH, CN, CO, OH, and TiO.

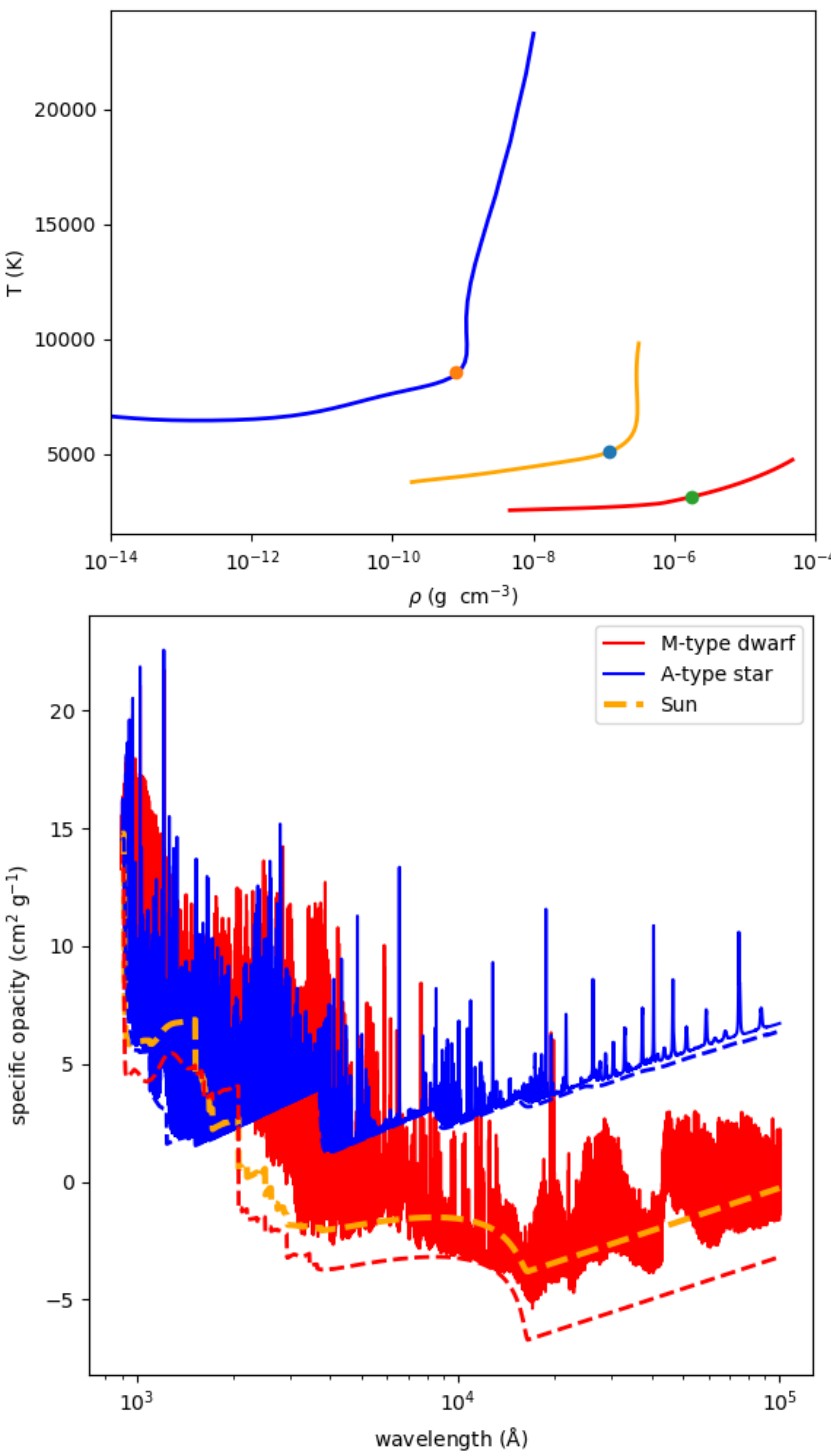

**Figure 3.** Similar to Figure 1 for the main sequence M-type ($T_{\text{eff}}$ = 3500 K) and A-type stars ($T_{\text{eff}}$ = 9800). The continuum opacity is shown with dashed lines. The total (including lines) opacity is omitted for the solar case since it is already shown in Figure 1. The representative data correspond to ($\rho \simeq 0.8 \times 10^{-9}$ g cm$^{-3}$, $T \simeq 8600$ K) in the case of the A-type star and ($\rho \simeq 1.8 \times 10^{-6}$ g cm$^{-3}$, $T \simeq 3200$ K) for the M-type star.

## 4. Observations

A sanity check is always necessary, for which we may use the solar spectrum. Nevertheless, it turns out that performing observations of the solar absolute flux poses notable difficulties related to its overwhelming brightness, large angular size on the sky, and the hurdles to calibrate the observations using laboratory sources or sky sources such as

standard stars. This is not to say that such observations have not been made. They, in fact, keep being made with regularity, motivated by the need to understand the amount and variability of the solar radiation impacting the Earth (see, e.g., [23,24]), but to our knowledge, there is no solar spectrum of reference available with a reliable calibration over a very broad spectral range.

An alternative is offered by the various solar analogs identified over the years, largely with the motivation of performing differential determinations of chemical abundances, which benefit from the cancelation of systematic errors inherent to that type of analysis. The star 18 Sco is the nearest such star available and has been studied in detail. Broad-coverage spectroscopy from the Hubble Space Telescope is available for this target, with high-quality absolute fluxes (see, e.g., [25]).

This star is close enough and bright enough that it has been observed with an interferometer, and its angular diameter has been resolved and measured to be $\theta = 0.6759 \pm 0.0062$ milliarcseconds [26]. The Gaia parallax for this star is $70.7371 \pm 0.0631$ milliarcseconds, implying it is at a distance $d$ of $14.14 \pm 0.01$ parsecs, and it has a radius of $R = \theta/2 \times d = 1.027 \pm 0.001 R_\odot$. The precision of the angular diameter, good to 1%, allows us to scale the model fluxes, computed at the stellar surface, and compare them to the flux measured for the star. Figure 4 illustrates this exercise, using a solar model ($T_{\text{eff}} = 5777$ K, $\log g = 4.437$, and the chemical composition from [19], [Fe/H]= 0—very close to the parameters for 18 Sco published from [26]: $T_{\text{eff}} = 5817 \pm 4$ K, $\log g = 4.448 \pm 0.012$, and [Fe/H] $= 0.052 \pm 0.005$), showing excellent agreement, except at the shortest wavelengths ($\lambda < 2300$), where multiple assumptions built in the models break down.

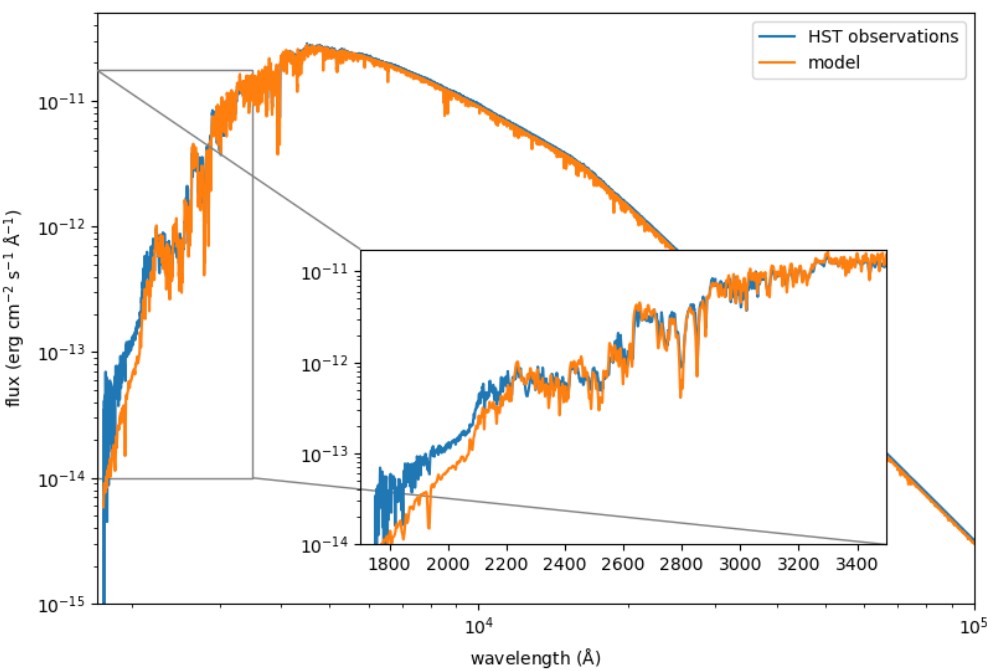

**Figure 4.** Observed (blue) and model (orange) spectra for the solar twin 18 Sco. The model has been scaled using the determination of the star's angular diameter from [26].

The light at wavelengths shorter than about 2000 Å escapes from very high atmospheric layers, where the classical model atmospheres we have adopted are no longer realistic due mainly to departures from Local Thermodynamical Equilibrium, the breakdown of the assumption of hydrostatic equilibrium, and the relevance of magnetic fields, which are ignored. An ad hoc chromospheric temperature increase was thought in the 1970s to solve the flux discrepancies found for the Sun at short wavelengths [27,28], but hydrostatic models cannot explain many of the observations such as the strength of CO bands [29].

A couple of features are noteworthy in Figure 4. At about 2500 Å a break is apparent, and a second one is visible near 2100 Å. These sudden reductions in flux correspond to the abrupt increases in opacity due to atomic magnesium and atomic aluminum, respectively. Other features from photoionization edges one may expect based on Figure 2 are not really visible, most likely, as pointed out above, due to the opacity enhancement shifting the region from which the continuum flux escapes to layers higher up in the atmosphere. The continuum changes in slope at about 4000 Å, longwards of which $H^-$ photoionization dominates, and 16,000 Å, where $H^-$ bremsstrahlung becomes the main contributor to the continuum opacity.

The brightest star in the sky, Sirius, is a good example of an A-type star, and its spectrum is included among the high-quality observations from the Hubble Space Telescope [30]. The star has a white dwarf companion that is irrelevant to our discussion. Its parallax is 379.21 ± 1.58 milliarseconds, or a distance of 2.637 pc. The angular diameter has been measured by [31] to be $\theta = 6.039 \pm 0.019$ milliarcsends, and more recently confirmed by ([32] as $\theta = 6.041 \pm 0.017$ milliarconds). Figure 5 shows the observed spectrum (blue) confronted with a model (orange) with $T_{\text{eff}} = 10{,}000$ K, $\log g = 4.0$ and $(\text{Fe/H}) = 0$ ([32] adopted for this star $T_{\text{eff}} = 9845$ K, $\log g = 4.25$ and $[\text{Fe/H}] = +0.5$, where the two latter parameters are inherited from [33]).

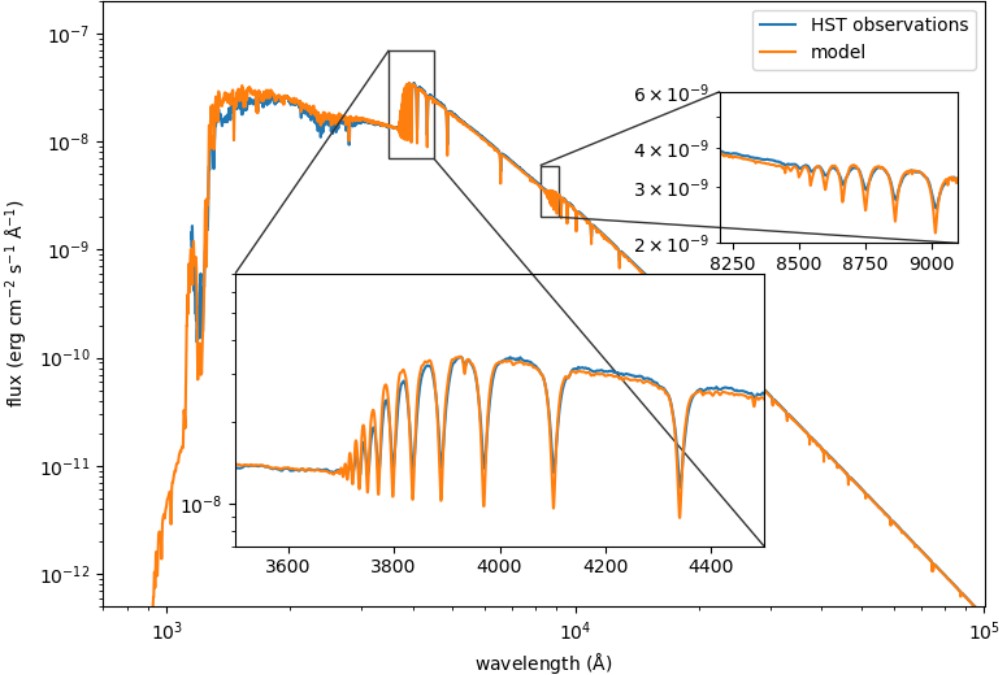

**Figure 5.** Observed (blue) and model (orange) spectra for the A-type Sirius A. The model has been scaled using the determination of the star angular diameter by [32].

The jumps in flux due to the photoionization of hydrogen are quite obvious at about 900 Å ($n = 1$), 3700 Å ($n = 2$), and 8200 Å ($n = 3$), becoming progressively weaker. The drop in flux at about 1300 Å is due to carbon photoionization, and there is a smaller drop at about 1500 Å caused by Si photoionization. The very strong lines at about 1200 Å are a blend of L$\alpha$ (H $n = 2$ to $n = 1$ transition) with S I lines on the blue side. The cores of the H lines are deeper in the models than in the data, which could be a limitation in the models, although departures from Local Thermodynamic Equilibrium work in the opposite sense and can therefore be excluded (e.g., [34]), or an issue with scattered light in the observations.

As an example of a cool star, we have chosen Gliese 555, a well-studied red dwarf star with an effective temperature of about 3200 K. This star is one of the few M dwarfs included in the Hubble Space Telescope sample with accurate fluxes, but unfortunately is not among

the short list of M dwarfs with measured angular diameters. Nonetheless, Ref. [35] have built a relation between the infrared luminosity of M dwarfs and their radii, using the stars with interferometric angular diameters, and arrived at $R = 0.310 \pm 0.013 R_\odot$ for GJ 555, which, combined with the Gaia parallax, leads to $\theta = 0.461 \pm 0.019$ milliarcseconds.

Figure 6 compares the observations with a model for $T_{\mathrm{eff}} = 3200$ K, $\log g = 5$ and [Fe/H]= 0 ([35] give $T_{\mathrm{eff}} = 3211$ K, $\log g = 4.89$ and [Fe/H]= +0.17), showing fair agreement. The parameters appear to be appropriate, and so is the angular diameter, but the model's imperfections are much more significant than in the cases of 18 Sco and Sirius. The complexity of the model is significantly higher due to the pervasive presence of molecules in the atmosphere of this star. As one would expect from the analysis in Section 3, and in particular from Figure 3, the shape of the spectra of this type of star is dominated by the presence of molecular bands of MgH, TiO, VO, and CaH, as well as very strong lines from low-lying levels of Na I (5900 Å and 8190 Å), K I (7680 Å), and Ca I (4227 Å).

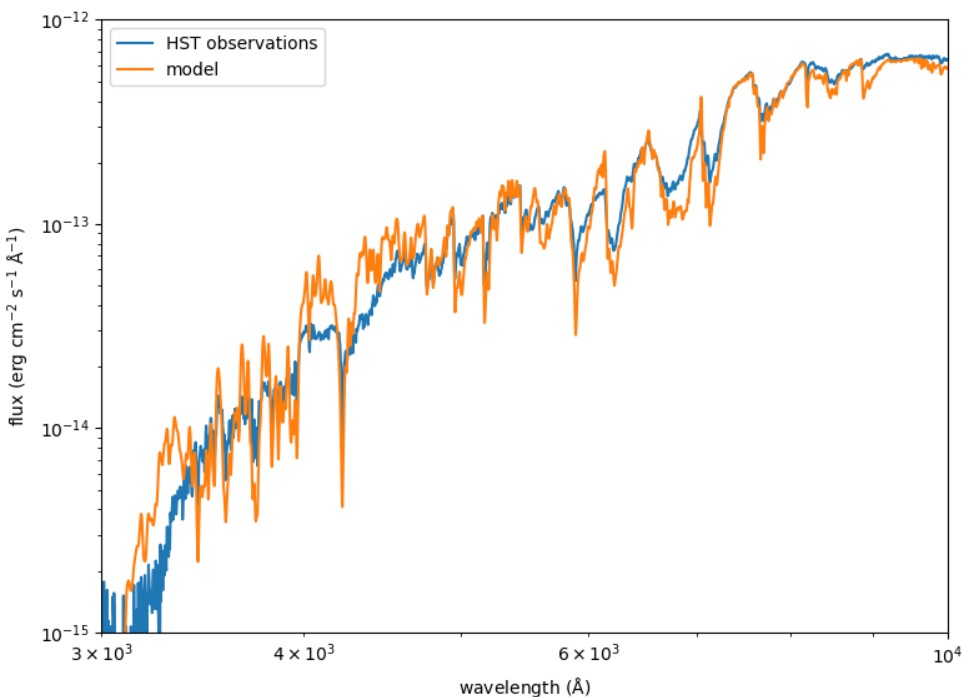

**Figure 6.** Observed (blue) and model (orange) spectra for the M-type dwarf star GJ 555. The model has been scaled using the estimated stellar radius from [35] and the Gaia DR3 parallax for the star.

## 5. Diagnostics

Most relevant for the study of stars and understanding their properties is how opacity in general and photoionization in particular vary depending upon the properties of the star.

The fundamental stellar parameters are mass and radius, plus age and chemical composition. To the zeroth-order, mass determines the fate of a star, including how long its life will be, what chemical elements it will be able to produce in its interior by nuclear fusion, and how it will die, with the most typical outcomes being a core-collapse supernova leaving behind a black hole or a neutron star or nothing for $M > 8 M_\odot$, or a white dwarf for lower-mass stars. Nevertheless, from the point of view of the spectrum of a star, the most relevant parameters are the star's surface (or *effective*) temperature, its surface gravity, and its chemical composition.

In the longest phase of the life of a star, the *main sequence*, it fuses hydrogen in its core to produce helium. In this period, the mass correlates perfectly with surface temperature: the more massive the star, the warmer the surface temperature. This is therefore the main atmospheric parameter that controls how the spectrum of the star looks, as illustrated in Figures 4–6. we now examine the impact of the other two parameters on the structure of the stellar atmosphere and the shape of the spectrum.

I have already discussed the situation for cool, M-type stars, where $H^-$ photoionization is still the main contributor to the continuum opacity in the optical and near-infrared, but the line absorption due to molecules dominates the opacity. In what follows, we will discuss the other two warmer cases considered in our previous examples: a solar-like star and an A-type star.

In the top panel of Figure 7, we can see the run of the main thermodynamical quantities for a solar-like star with Rosseland optical depth, which is a weighted mean of the integrated opacity along the atmosphere down to a given depth, and gives a very useful reference axis when studying optical properties. There are three models shown in the figure: a reference solar-like model (in blue), another with 0.5 dex higher gravity (orange), and a third, which, in addition to the higher gravity, has a higher metal content ([Fe/H] = +0.5). The bottom panel of the figure shows the corresponding model spectra.

Our models assume that the atmosphere is in hydrostatic equilibrium

$$\frac{dP}{dm} = g, \tag{1}$$

where $P$ is the gas pressure (turbulent and radiation pressure are negligible for this type of atmosphere), $m$ is the mass colum, and $g$ is the gravitational acceleration $g = GM/R^2$. An increase in gravity compresses the atmosphere, enhancing the pressure, while keeping the fractional contribution from electrons ($Pe$) to it at a similar level ($10^{-4}$, except in the deepest layers where H begins to be ionized). The effect of this change on the continuum opacity is negligible, since the abundance of $H^-$, proportional to $Pe$, increases only mildy, while $Pe/P$ stays nearly constant.

On the other hand, an additional increase in the abundance of the heavy elements has a profound impact on the near-UV opacity, due to the importance of iron and magnesium photoionization, and the increase in iron (and other metal) lines, and the UV flux is consequently reduced. The enhanced UV opacity cools down the outer atmospheric layers and heats the deepest ones (an effect known as *backwarming*). There is a small increase in the electron pressure, partly from the increase in the abundance of electron donors such as sodium, magnesium, calcium, and iron, which would in principle boost the optical and near-IR opacity, therefore reducing the flux at those wavelengths, but the effect observed is exactly the opposite. This is in part due to the fact that the relative enhancement of electron pressure in high atmospheric layers disappears when reaching the continuum-forming layers at the Rosseland optical depth near unity. Furthermore, an increase in continuum opacity does not necessarily imply a reduction in flux, since the model needs to self-adjust to satisfy energy conservation, which for these three models imposes that the flux integrated over all wavelengths must be the same

$$\int_0^\infty F_\lambda d\lambda = F = \sigma T_{\text{eff}}^4, \tag{2}$$

and therefore a decrease in the UV flux has to be compensated at other wavelengths.

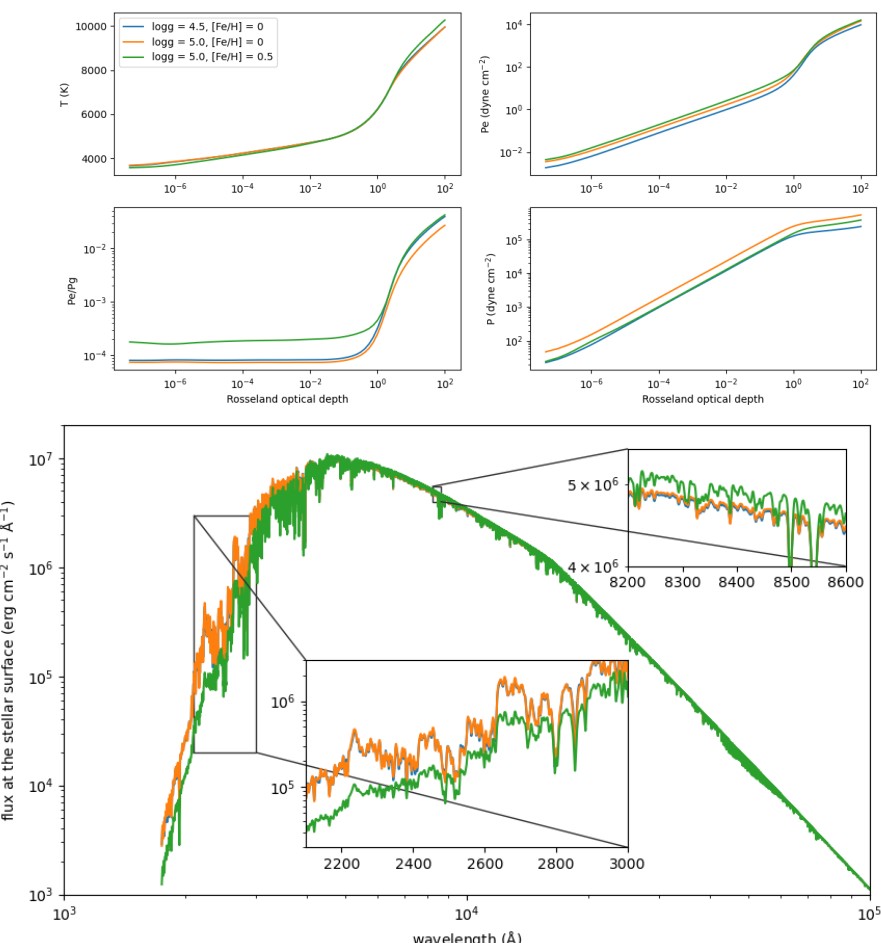

**Figure 7.** (**Upper panel**) Run of various thermodynamical quantities (clockwise: temperature, electron pressure (*Pe*), gas pressure (*P*) and their ratio (*Pe/P*)) with the Rosseland mean opacity for model atmospheres for a solar-like star ($T_{\mathrm{eff}}$ = 5777 K). (**Lower panel**) Predicted emergent flux at the atmospheric surface for the models in the upper panels (no scaling is necessary, since we are only comparing models).

The situation for an A-type star such as Sirius, illustrated in Figure 8, is different. Here, H$^-$ plays only a minor role in the continuum opacity, and hydrogen atoms are the main contributors in the optical and near-infrared regions. An increase in surface gravity leaves the run of temperature with optical depth unchanged but compresses the atmosphere, enhancing the gas pressure and, to a lesser extent, the electron density, with an overall reduction in the electron's partial pressure. An additional increase in the abundances of the heavy elements does not change the atmospheric structure much.

In the lower panel of Figure 8, we can appreciate how the changes described affect the emergent radiative flux. The boost in pressure associated with the increase in surface gravity broadens the lines somewhat , both the hydrogen lines and other strong features. The pressure enhancement and the subsequent reduction in the electron partial pressure increases the ionization and dampens slightly the hydrogen photoionization, as clearly visible in the Balmer (3700 Å) and the Paschem (8500 Å) jumps. An enhancement in the metal abundance noticeably affects the UV absorption, in this case mainly due to (ionized) iron photoionization, reducing the UV flux, which is compensated with a slightly increase at other wavelengths to keep the integrated flux constant.

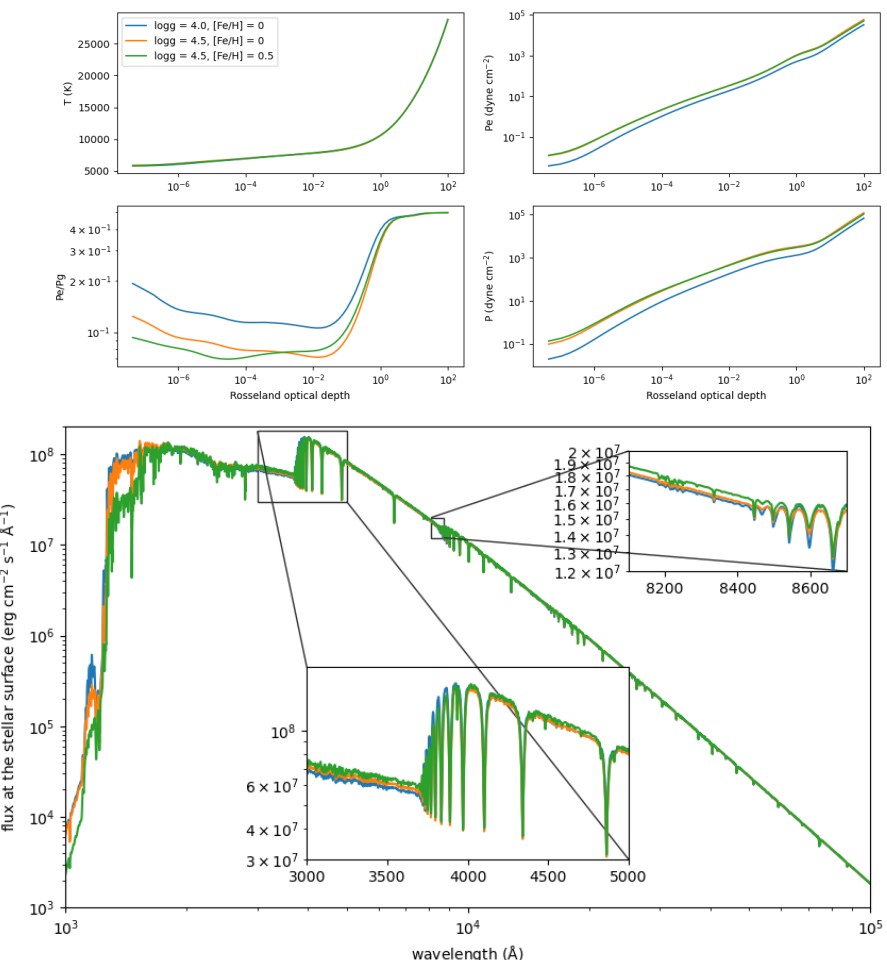

**Figure 8.** (**Upper panel**) Run of various thermodynamical quantities (clockwise: temperature, electron pressure ($Pe$), gas pressure ($P$), and their ratio ($Pe/P$)) with the Rosseland mean opacity for model atmospheres for an A-type star ($T_{\text{eff}}$ = 9750 K). (**Lower panel**) Predicted emergent flux at the atmospheric surface for the models in the upper panels.

## 6. Summary and Conclusions

We have used a state-of-the-art code for computing synthetic spectra and standard plane-parallel model atmospheres to evaluate the role of photoionization in shaping the spectral energy distributions of stars.

The photoionization of atomic hydrogen or the $H^{-}$ ion are a dominant source of opacity in the optical and infrared regions in stars with one solar mass or more. While $H^{-}$ remains a dominant contributor to the optical/infrared continuum opacity, molecular line opacity becomes more important in cooler (usually less massive) stars and causes a significant redistribution of the emergent flux. This makes it harder to model stellar spectra of M-type dwarfs, which are the most common stars across the Milky Way, than warmer stars.

The ultraviolet spectra of most stars are dominated by photoionization from heavier elements (magnesium, aluminum, silicon, and iron), as well as atomic line absorption. In models of solar-type stars, the ultraviolet opacity seems to be accounted for appropriately, although a more exhaustive study is necessary to make sure that (a) no important contributors are missed (e.g., photoionization from elements with similar atomic mass or heavier than iron), and (b) no opacity sources are counted twice, (e.g., autoionization lines being included in the atomic line list and in the photoionization cross-sections).

The illustrative examples given in this paper can serve as a starting point for new, deeper, investigations, looking at the sources of opacity in various types of stars. Improving our understanding of the atmospheric opacity paves the way to refining the agreement between the observed spectral energy distributions of stars and model predictions, which are key to our ability to infer stellar properties, such as mass, radius, luminosity, chemical composition, etc., from observations.

**Funding:** The author acknowledges support for this research from the Spanish Ministry of Science and Innovation (MICINN) projects s AYA2017-86389-P and PID2020-117493GBI00. Funding for the DPAC has been provided by national institutions, in particular the institutions participating in the *Gaia* Multilateral Agreement.

**Institutional Review Board Statement:** Not applicable.

**Informed Consent Statement:** Not applicable.

**Data Availability Statement:** The opacity tables and the model spectra used in this paper have been computed with version v1.2 of Synple, publicly available from github.com/callendeprieto/synple, accessed on 13 January 2023.

**Acknowledgments:** I am thankful to Ivan Hubeny for his comments on an early draft of this manuscript. This work has made use of data from the European Space Agency (ESA) mission *Gaia* (https://www.cosmos.esa.int/gaia, accessed on 16 January 2023), processed by the *Gaia* Data Processing and Analysis Consortium (DPAC, https://www.cosmos.esa.int/web/gaia/dpac/consortium, accessed on 23 January 2023). This research has made use of the SIMBAD database, operated at CDS, Strasbourg, France.

**Conflicts of Interest:** The author declares no conflict of interest.

## Notes

[1] The usual convention in astronomy is used for this parameter, which prescribes that all the elements heavier than He are changed in the same ratio relative to their solar abundances, and that ratio is quantified with the parameter $[\text{Fe/H}] = \log\left(\frac{N_{Fe}}{N_H}\right) - \log\left(\frac{N_{Fe}}{N_H}\right)_\odot$, where $N_X$ is the number density for the element $X$.

[2] kurucz.harvard.edu accessed on 25 October 2022, file gfall08oct17.dat.

[3] github.com/callendeprieto/synple accessed on 13 January 2023.

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
