# Peer review of "The Shapes of Stellar Spectra"

_atoms, doi:10.3390/atoms11030061_

Round 1

Reviewer 1 Report

This manuscript shows the stellar energy distribution to examine the role of photoionization for solar-like, A-type, and M-type stars. A plane-parallel model atmosphere and a spectral synthesis code, adopting the updated atomic data, were used to calculate the stellar wavelength-dependent flux with various temperatures, surface gravities, and abundances. The calculations show that photoionization of H I and H is the main contributor to the continuum opacity in the optical and near-infrared, and the contributions of heavier elements such as Mg, Al, Si, Fe, are important in the ultraviolet.

The study on the effects of the photoionization process on stellar continua is very interesting for us to understand the spectra with different temperatures. The present manuscript is important and deserves to be published on Atoms. I suggest the author revise the present manuscript, and improve the text further, especially,

1. When showing the opacity at a fixed temperature and density, such as in Fig.1, how did the author decide from where the optical continuum flux is escaping?

2. Compared with HST observations of solar-like star 18 Sco, the model gives lower flux than the observations in the ultraviolet. Does it mean the theoretical opacity is not missing, but too high? Which element(s) is responsible for such excess of opacity?

3. For Fig. 4 and Fig. 7, the scales of flux are 1014 1011 and 103 107 for a solar-like star respectively. It is better to give more details on scaling the model fluxes to the observed.

4. The references of the atmospheric parameters for 18 Sco, Sirius, and GJ555 should be added.

Author Response

My sincere thanks to the referee for his careful work that has improved the paper. My reply is included below.

1. When showing the opacity at a fixed temperature and density, such as in Fig.1, how did the author decide from where the optical continuum flux is escaping?

Since the qualitative results and the conclusions about the relevant contributors will not depend very much on the exact parameters, this was done approximately by choosing an atmospheric layer from the models slightly higher than optical depth unity. A note to clarify this has been added to the caption of Figure 1. We have as well made it clear the values of the parameters adopted in all cases/figures.

2. Compared with HST observations of solar-like star 18 Sco, the model gives lower flux than the observations in the ultraviolet. Does it mean the theoretical opacity is not missing, but too high? Which element(s) is responsible for such excess of opacity?

The light at wavelengths shorter than about 2000 AA escapes from very high atmospheric layers where the model atmospheres are no longer realistic, due mainly to departures from Local Thermodynamical Equilibrium, the breakdown of the assumption of hydrostatic equilibrium and the relevance of magnetic fields, which are ignored. A chromospheric temperature increase was found in the 1970's to solve the flux discrepancies found for the Sun at short wavelengths, but hydrostatic models cannot explain many of the observations such as the strength of various molecular bands. This explanation has been added to the text, together with appropriate references.

3. For Fig. 4 and Fig. 7, the scales of flux are 10−14 − 10−11 and 103 − 107 for a solar-like star respectively. It is better to give more details on scaling the model fluxes to the observed.

In Figs. 4-5-6 the fluxes are as observed at Earth, and the models have been scaled, while in Figs. 7-8 we compare only models and therefore the fluxes are at the stellar surface to avoid any uncertainty in the transformation. We have made this clear in the figure caption for Fig. 7.

4. The references of the atmospheric parameters for 18 Sco, Sirius, and GJ555 should be added.

done

Reviewer 2 Report

This manuscript is suited as a first introduction to, or an overview of, the physics involved in the formation of a stellar spectrum.  It is pedagogically written for students or researchers in the general field of physics but offers no new insights to the research field.

My comments mainly concern details and some are merely suggestions:

Abstract: One could mention that also spectral line absorption is important for the spectra of cool stars.

Introduction, line 31: The five wavelengths stated are far from the five first nominal absorption edges of neutral hydrogen: 91.2, 364.7, 820.6, 1458.8, and 3282.3 nm.

Page 2, line 79: An extra "k" appears in the molecule list between AlO and CaH

Fig. 1, 2, and 3 labels: Suggest using "g" rather than "gr" for the cgs mass unit, alternatively SI units. In the field of stellar atmosphere research cgs units are often used. As this manuscript appears to be directed to a more general audience why one might consider using SI units throughout

Page 4, line 96: "The top panel of 1 illu..." -> "The top panel of Fig. 1 illu..."

Page 5, Fig. 2 label: "cn2" -> "cm2"

Fig. 3, caption, two last lines: Units for density are missing

Page 7, lines 130-131: (Molecular) line absorption severely blocks much of the continuum flux (as is actually acknowledged further down)

Page 8, line 168: "about 2500" -> "about 2600" in the figure

Page 8, line 185-186: There is no hydrogen absorption edge at 1400 A. Rather some metallic edge?

Fig. 7 and 8: The axis and descriptive labels should not be smaller than the general paper text, perhaps the figures can fill the full text width? 

Page 12, line 272: "Pashem" -> "Paschen"

I suffer from a (particularly among men) common "colour blindness" and find that the colours are well chosen (except in Fig. 2 which is difficult, perhaps more line styles may be used).  The three colours blue, orange, and black are supposedly the best choices for various flavours of colour blindness

The readability of the manuscript would fare well by some English language editing

I do not find it necessary to se the revised version

Author Response

Thank you so much to the referee for his useful comments and suggestions, which have improved the paper. My reply is included below.

Abstract: One could mention that also spectral line absorption is important for the spectra of cool stars.

done

Introduction, line 31: The five wavelengths stated are far from the five first nominal absorption edges of neutral hydrogen: 91.2, 364.7, 820.6, 1458.8, and 3282.3 nm.

yes, sorry, these were eyeballed in the first draft and I forgot to correct them later. They have been updated now.

Page 2, line 79: An extra "k" appears in the molecule list between AlO and CaH

fixed

Fig. 1, 2, and 3 labels: Suggest using "g" rather than "gr" for the cgs mass unit, alternatively SI units. In the field of stellar atmosphere research cgs units are often used. As this manuscript appears to be directed to a more general audience why one might consider using SI units throughout

Changed gr to g throughout. I did consider changing all the units to SI early on but thought that losing the ability to make quick comparisons with the literature outweighted the advantages.

Page 4, line 96: "The top panel of 1 illu..." -> "The top panel of Fig. 1 illu..."

fixed

Page 5, Fig. 2 label: "cn2" -> "cm2"

fixed

Fig. 3, caption, two last lines: Units for density are missing

fixed. I also found a few other occurrences that have been fixed as well.

Page 7, lines 130-131: (Molecular) line absorption severely blocks much of the continuum flux (as is actually acknowledged further down)

added

Page 8, line 168: "about 2500" -> "about 2600" in the figure

There are multiple discontinuities and I want to highlight the one at 2500 A since we can easily identify it corresponds to an edge of atomic Mg.

Page 8, line 185-186: There is no hydrogen absorption edge at 1400 A. Rather some metallic edge?

Yes, close inspection revealed there are two jumpts, one at 1300 and another at 1500 A, due to 
carbon and silicon atoms, respectively. The text has been corrected accordingly.

Fig. 7 and 8: The axis and descriptive labels should not be smaller than the general paper text, perhaps the figures can fill the full text width? 

done

Page 12, line 272: "Pashem" -> "Paschen"

fixed